# Genetic Analysis of Fruit Traits in Wolfberry (*Lycium* L.) by the Major Gene Plus Polygene Model

Xiaoyue Ren [1,†], Haoxia Li [2,†], Yue Yin [1], Linyuan Duan [1], Yajun Wang [1], Xiaojie Liang [1], Ru Wan [1], Ting Huang [1], Bo Zhang [1], Wanpeng Xi [3], Wei An [1,*] and Jianhua Zhao [1,*]

[1] National Wolfberry Engineering Research Center, Wolfberry Science Research Institute, Ningxia Academy of Agriculture and Forestry Sciences, Yinchuan 750002, China; rxy_2008@126.com (X.R.); yueyin0112@aliyun.com (Y.Y.); dly698013@163.com (L.D.); yajun@163.com (Y.W.); lxj910303@126.com (X.L.); wanru2008@163.com (R.W.); ht20180312@163.com (T.H.); zhang_bo_0309@126.com (B.Z.)

[2] Desertification Control Research Institute, Ningxia Academy of Agriculture and Forestry Sciences, Yinchuan 750002, China; lihaoxia0943@163.com

[3] College of Horticulture and Landscape Architecture, Southwest University, Chongqing 400716, China; xwp1999@zju.edu.cn

[*] Correspondence: angouqi@163.com (W.A.); zhaojianhua0943@163.com (J.Z.)

[†] These authors contributed equally to this work.

**Abstract:** The fruit diameter (FD), fruit length (FL), fruit peduncle length (FPL), fruit weight (FW) and fruit index (FI, FL/FD) are important quantitative traits in wolfberry fruit, and also one of the most important goals of variety breeding; however, the inheritance of these traits has not been studied to date. In this study, the genetic analysis of these five fruit traits was undertaken for four pairs of $F_1$ hybrid populations (CI, CII, CIII and CIV) using the major gene and polygene mixed inheritance model. The results showed that the five fruit traits exhibited super-parent segregation in four hybrid combinations, and five traits of progeny with abundant genetic diversity. In CII, CIII and CIV, the mid-parental heterosis ratio ($RH_m\%$) of FD, FL, FPL and FI was greater than 0 with positive heterosis. FD, FL and FI in CI, CII and CIII were controlled by one pair of additive-dominant major genes (A-1). However, in CIV, FD was controlled by two pairs of additive-dominant alleles (B-6) and FL was best fitted to polygenic control (A-0). In addition, it was found that FPL in CI, CIII and CIV was controlled by one or two pairs of additive-dominant major genes (A-1, B-6, B-1), and FW in CIII and CIV was also controlled by one or two pairs of additive-dominant major gene controls (A-1, B-1). For FD, FPL, FW and FI in CIII and FPL and FW in CII, the major genes heritability was over 50%, indicating that these traits are affected by both genes and the environment, and that the selection of these traits should be considered in later generations due to the large effect of environmental factors. Therefore, this study provides a theoretical basis for QTL mapping and early selection of hybrid breeding of *Lycium* fruits.

**Keywords:** wolfberry; fruit traits; genetic analysis; major gene plus polygene

## 1. Introduction

The inheritance system of quantitative traits can consist both of a few major genes and a number of polygenes. One widely utilized model is called the mixed major gene plus polygene inheritance model (mixed-inheritance model or mixed-genetic model) [1]. In recent years, the technology for the analysis of plant quantitative traits by the mixed-inheritance model has been developed [2], allowing the detection of the effects of quantitative trait major genes and polygenes and the estimation of their genetic parameters, usually validated and supplemented quantitative trait loci (QTL) [3]. A total of seven types of genetic models, including one pair of major genes (code A), two pairs of major genes (code B), polygenes (code C), one pair of major genes + polygenes (code D), two pairs of major genes + polygenes (code E), three pairs of major genes (code F) and three pairs of major genes

+ polygenes (code G), have been established to quantitate the genetic and environment effects [2]. On the basis of gene additive effect, each type of model is numbered according to whether there is a dominant effect, positive dominance, negative dominance, complete dominance or partial dominance. Mixed inheritance analysis for wheat flag leaf area (FLA) has been reported using joint segregation analysis of crosses in different years. The results indicated that the FLA was mostly under control of one major gene in combination with polygenes (model D-2) during the first year; however, it was controlled by mixed epistasis of two major genes plus polygenes (model E-1) during the second year [4]. This method has also been applied to rice crop. Zheng et al. [5] analyzed the optimal model for stripe disease resistance in the rice restorer line, C224, also called E-1. Similarly, this method can also be used to identify the existence of major genes and polygenes for agronomic traits in vegetables.

The time to flowering in chickpea was controlled by two major genes along with other polygenes [6]. The number of kernels per horn of *Brassica napus L.* seeds was best described by the E-0 inheritance model, a case of two additive-dominance-epistasis major genes as well as additive-dominance-epistasis polygenes [7]. The traits of tiller number and leaf number in non-heading Chinese cabbage were controlled by two major genes and polygenes with dominant, additive and epistasis effects, respectively [8], while the tomato internode length was mainly controlled by major gene and should be selected in early populations [9]. In addition, Qi et al. [10] analyzed the fruit cracking of melon through a mixed major gene plus polygene inheritance model and found that the inheritance model E-0 for resistance, incorporated two additive-dominance-epistasis major genes plus an additive-dominance-epistasis polygene.

The mixed inheritance model has also been applied to horticultural plants. The internode length (IL) in crape myrtle fit the D-0 model, while the plant height and primary lateral branch height were quantitative traits fitting the E-0 model, namely two additive-dominance-epistasis major genes plus additive-dominance-epistasis polygenes [11]. Both the corolla tube merged degree and the relative number of ray florets in chrysanthemum could be described by a B-2 inheritance model via two additive-dominance major genes [12]. The plant height, individual flower height and flower diameter traits in the ornamental, bearded iris (*Iris germanica*) were all in accordance with the additive-dominance-epistasis major gene plus additive-dominance polygene inheritance model (E-1) [13].

This mixed inheritance method has also been successfully applied to the genetic analysis of hybrid plants, such as hybrids from *Plumbago auriculata* and is variant, *Plumbago auriculata* f. alba. The main flower characteristics, including the number of flowers per inflorescence, beginning of the blooming period, flower length and flower diameter, were all controlled by two pairs of major genes, while inflorescence length and inflorescence diameter were best fitted by the A-0 model, indicating that both characteristics were controlled by only polygenes. In addition, the inflorescence number was best modeled by A-1, which was controlled by a pair of additive-dominant-epistatic major genes [14].

Wolfberry is a perennial deciduous shrub of the family Solanaceae (*Lycium* L.), consisting of about 80 species worldwide. Wolfberry is a shrub native of China, where there are seven species and three varieties of in the natural distribution [15]. Wolfberry fruit is an important edible and source for traditional medicine., *L. barbarum* fruit polysaccharides have anti-oxidant, anti-tumor, anti-aging and immunity enhancement benefits in the human body [16]. The major fruit alkaloid, betaine, has various pharmacological effects, such as promoting fat metabolism and anti-fatty liver and kidney protection [17]. Carotenoids are not only the main pigment substances in wolfberry fruits, but also one of its active medicinal ingredients, with anti-cancer and anti-aging effects as well as enhancing immune functions and preventing atherosclerosis [18]. The fruit traits provide important assessment standards for quality evaluation, classification and grading in the current wolfberry market [19]. Most fruit indicators provide quantitative traits, which are generally regulated by micro-effect polygenes with complex genetic regulatory mechanisms that are easily influenced by environmental conditions. Consequently, for most fruit traits,

there is no clear correspondence between phenotype and genotype [20]. Wolfberry hybrid breeding programs were initiated relatively recently, and as a consequence, the research into the inheritance of its fruit traits in hybrid offspring is lagging behind other commercial crops [21]. Furthermore, the main gene plus polygene inheritance model is yet to be applied to wolfberry.

In this study, the mixed inheritance model was used to analyze fruit-related traits in the $F_1$ population in wolfberry for two consecutive years, and to identify the genetic rule in wolfberry fruits. This research provides a foundation for clarifying the genetics underlying wolfberry fruit qualities and a reference for molecular marker assisted breeding of wolfberry.

## 2. Materials and Methods

### 2.1. Plant Material

The test materials were obtained from the National wolfberry Engineering Research Center (38°380′ N, 106°9′ E, 1100 m above sea level) and were grown at the wolfberry test base in Luhua Town, Xixia District, Yinchuan City, Ningxia. In cross I, the female parent 'Ningqi No. 1' (NQ1) was an artificial cultivar, with wide applicability, stable yield, the largest artificial planting area and red and oval fruit, and the male parent 'Ningxia Huang-guo' (NH) was orange-yellow and spindle-shaped; 228 hybrid offspring were obtained. In Cross II, the female parent was "NQ1", and the male parent, 'Yunnan gouqi' (YN), was yellow-red and spherical; 363 hybrid offspring were obtained. In Cross III, the female parent, 'Baihua gouqi' (BH), was red and spindle-shaped, and the male parent, 'Chinese gouqi' (ZG), was red and oblong; 131 hybrid offspring were obtained. In Cross IV, the female parent was 'ZG' and the male parent was 'NH'; 268 hybrid offspring were obtained. A total of 1184 F1 plants were obtained. The hybrid seeds were colonized in 2014. After the growth was stable, the related traits of the fruit of the four parents and the F1 populations were investigated in 2020 and 2021, respectively (Figure 1).

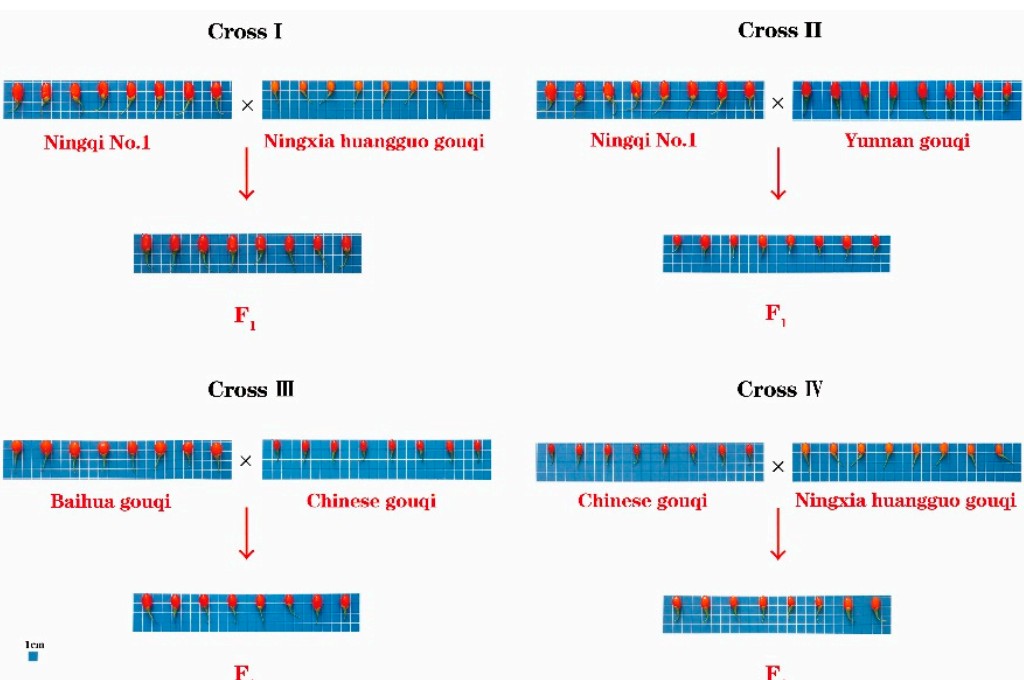

**Figure 1.** Fruit morphology of four hybrid parents.

### 2.2. Phenotypic Data Collection

The fruit diameter/mm (FD), fruit length/mm (FL), fruit peduncle length/mm (FPL), fruit weight/g (FW) and fruit index (FI) were measured according to the methods of Shi et al. [22]. Briefly, FW is counted as the weight of one mature fruit, and FL is the

maximum distance between fruits top to bottom. FD is the widest distance across a fruit. FPL is the length of the fruit stalk measured with a vernier caliper, and the average value is obtained. The FI is the ratio of the FL to the FD (FI = FL/FD).

### 2.3. Statistical Analysis and Heterosis Analysis

The minimum value, maximum value, mean value, standard deviation, skewness, kurtosis and coefficient of variation (C.V.% = standard deviation/mean) of wolfberry fruit traits were calculated. The main fruit traits in $F_1$ hybrids from four crosses were plotted for frequency distribution during different years (2020–2021), with a joint analysis of variance. Pearson correlation analysis was used to analyze the relationship between the different traits. The heterosis of fruit traits was measured by the middle parent's value (MPV), the mid-parent heterosis ($H_m$), the ratio of mid-parent heterosis ($RH_m$) and the ratio of heterobeltiosis ($RH_b$) according to the methods of Gao et al. [23]. The above data analysis was calculated using Microsoft Excel 2010 (Microsoft Corporation, Redmond, WA, USA) and IBM SPSS Statistics 20.0 software (IBM, Almonk, NY, USA).

### 2.4. Mixed Inheritance Analyses

The $F_1$ wolfberry population was considered a pseudo-$F_2$ population for genetic analysis in accordance with the double-pseudo-testcross strategy. A single-generation segregation analysis as described by Gai et al. [2] was used to analyze the mixed inheritance model of wolfberry fruit traits, including the major inheritance model, polygene model and mixed model. This analysis method divided the phenotypic traits of parents and $F_1$ into different distribution intervals and counted the distribution times in each interval. The description for the genetic model of $F_2$ population are shown in Table 1. A total of 11 kinds of inheritance models were used to analyze the wolfberry $F_1$ fruit morphological data and evaluated according to the minimum Akaike information criterion (AIC) value. The selected model was then evaluated by a goodness-of-fit test based on five statistical parameters, including Uniformity tests ($U1^2$, $U2^2$, $U3^2$), Smirnov's statistics ($_nW^2$) and Kolmogorov's statistics ($D_n$). After the parameters corresponding to the optimal inheritance model are determined, the major gene variance, phenotypic variance and other relevant parameters were further estimated through the least square method of distributed parameters [2]. Finally, the major gene heritability $h^2_{mg} = \sigma^2_{mg}/\sigma^2_p$ (where $\sigma^2_{mg}$ is the major gene variance and $\sigma^2_p$ is the phenotypic variance) were calculated. The software for segregation analysis (SEA, Yuan-ming Zhang, Ningjing, China) was provided by Professor Yuan-ming Zhang, National Key Laboratory of Crop Genetics and Germplasm Enhancement, Soybean Research Institute, Nanjing Agricultural University, Nanjing, China.

**Table 1.** Descriptions of the genetic models.

| Model Implication | Model | Model Description |
|---|---|---|
| 0MG | A-0 | Polygenes |
| 1MG-AD | A-1 | A pair of additive-dominant major genes |
| 1MG-A | A-2 | A pair of additive major genes |
| 1MG-EAD | A-3 | A pair of fully dominant major genes |
| 1MG-AEND | A-4 | A pair of negative fully dominant major genes |
| 2MG-ADI | B-1 | Two pairs of additive-dominant-epistatic major genes |
| 2MG-AD | B-2 | Two pairs of additive-dominant major genes |
| 2MG-A | B-3 | Two pairs of additive major genes |
| 2MG-EA | B-4 | Two pairs of equally additive major genes |
| 2MG-AED | B-5 | Two pairs of fully dominant major genes |
| 2MG-EEAD | B-6 | Two pairs of equally dominant major genes. |

Note: MG represents the major inheritance model; A represents the additive effect; D represents the dominance effect; E represents equal; N represents negative; I represents the epistatic interaction.

## 3. Results

### 3.1. Statistical Analysis of Fruit Characteristics in Hybrid Offspring

All fruit traits exhibited extremely significant differences ($p < 0.01$) among the $F_1$ hybrids from four crosses (Table 2). However, FL, FW and FI also showed highly significant differences between 2020 and 2021, indicating that environment has a stronger influence on these traits, and the traits are unstable. The variation coefficients of five fruit traits in 2 years ranged from 10.47% to 35.39% (Table 3). Among them, FL (14.81–17.08%), FPL (15.73–16.25%) and FW (35.14–35.49%) were all greater than 15%, which further indicated that these quantitative traits were largely separated in hybrid offspring. In addition, it was also found that the variation coefficient of FD, FL, FPL and FI gradually decreased with the increase of years. The decline also proved that these traits gradually stabilized with the growth of the year.

**Table 2.** Complex variance analysis of the main fruit characteristics in $F_1$ hybrids from four crosses.

| | | | Populations | | | | | | | |
|---|---|---|---|---|---|---|---|---|---|---|
| **Source** | | **Repetition** | **Cross I** | | **Cross II** | | **Cross III** | | **Cross IV** | |
| | | | **2020** | **2021** | **2020** | **2021** | **2020** | **2021** | **2020** | **2021** |
| FD | DF | 3 | 472 | 472 | 692 | 692 | 462 | 462 | 274 | 274 |
| | MS/Mean | 177.90 | 7.74 | 8.40 | 9.28 | 9.16 | 8.23 | 8.18 | 8.79 | 9.16 |
| | F /*p*-values | 233.78 ** | | 0.000 ** | 0.081 | | 0.461 | | | 0.000 ** |
| FL | DF | 3 | 472 | 472 | 692 | 692 | 462 | 462 | 274 | 274 |
| | MS/Mean | 1498.61 | 14.06 | 16.04 | 17.32 | 18.19 | 13.62 | 14.90 | 14.07 | 16.31 |
| | F /*p*-values | 315.818 ** | | 0.000 ** | | 0.000 ** | | 0.000 ** | | 0.000 ** |
| FPL | DF | 3 | 472 | 472 | 692 | 692 | 462 | 462 | 274 | 274 |
| | MS/Mean | 838.74 | 17.51 | 18.03 | 18.84 | 16.22 | 15.65 | 14.63 | 16.21 | 15.86 |
| | F /*p*-values | 131.98 ** | | 0.024 * | 0.000 ** | | 0.000 ** | | 0.189 | |
| FW | DF | 3 | 472 | 472 | 692 | 692 | 462 | 462 | 274 | 274 |
| | MS/Mean | 19.21 | 0.46 | 0.63 | 0.83 | 1.00 | 0.51 | 0.58 | 0.63 | 0.76 |
| | F /*p*-values | 511.64 ** | | 0.000 ** | | 0.000 ** | | 0.000 ** | | 0.000 ** |
| FI | DF | 3 | 472 | 472 | 692 | 692 | 462 | 462 | 274 | 274 |
| | MS/Mean | 6.20 | 1.83 | 1.94 | 1.88 | 2.01 | 1.66 | 1.83 | 1.61 | 1.79 |
| | F/*p*-values | 106.65 ** | | 0.000 ** | | 0.000 ** | | 0.000 ** | | 0.000 ** |

Note: * and **, 0.05 and 0.01 significance by one-way ANOVA and *t*-test, respectively. MS, mean square; DF, degrees of freedom.

**Table 3.** Representative values of the main fruit traits in $F_1$ hybrids from four crosses.

| **Traits** | **Year** | **Minimum** | **Maximum** | **Mean** | **Standard Deviation** | **C.V.%** |
|---|---|---|---|---|---|---|
| FD | 2020 | 5.67 | 11.99 | 8.57 | 1.10 | 12.83 |
| | 2021 | 6.13 | 11.80 | 8.72 | 0.91 | 10.47 |
| FL | 2020 | 9.50 | 24.94 | 15.14 | 2.59 | 17.08 |
| | 2021 | 7.77 | 25.75 | 16.56 | 2.45 | 14.81 |
| FPL | 2020 | 7.31 | 26.81 | 17.39 | 2.83 | 16.25 |
| | 2021 | 8.67 | 27.28 | 16.16 | 2.54 | 15.73 |
| FW | 2020 | 0.17 | 1.42 | 0.63 | 0.22 | 35.14 |
| | 2021 | 0.27 | 3.97 | 0.77 | 0.27 | 35.39 |
| FI | 2020 | 1.22 | 2.82 | 1.77 | 0.24 | 13.54 |
| | 2021 | 1.21 | 2.83 | 1.92 | 0.26 | 13.37 |

Note: C.V.%: coefficient of variation (C.V.% = standard deviation/mean).

### 3.2. Heterosis of Wolfberry Fruit Characteristics

Table 4 shows that in the CII, CIII and CIV F1 groups, FD, FL and FW have a ratio of mid-parent heterosis in $RH_m$% greater than zero, showing a positive advantage, with values ranging from 15.08–24.48%, 3.95–20.95% and 26.03–60.70%, respectively, indicating

that these traits of the three hybrid combinations have obvious heterosis. However, the FL, FPL, FW and FI in $F_1$ of CI showed negative dominance with a $RH_m$% was less than 0, indicating that there was a dominant genetic effect in the heterosis of these four fruit traits. At the same time, the RHm% values for FI in CII and CIV $F_1$ were also negative, which proved that the trait also had a dominant genetic effect in the heterosis.

**Table 4.** Heterosis of five fruit traits in $F_1$ hybrids from four crosses.

| Traits | Combination | Years | $P_1$ | $P_2$ | $F_1$ | Mid-Parent Value | Skewness | Kurtosis | $H_m$ | $RH_m$% | $RH_b$% |
|---|---|---|---|---|---|---|---|---|---|---|---|
| FD | Cross I | 2020 | 8.44 | 8.52 | 7.74 | 8.48 | 0.14 | 0.20 | −0.74 | −8.75 | −9.15 |
| | | 2021 | 7.88 | 7.70 | 8.40 | 7.79 | 0.52 | 0.62 | 0.61 | 7.80 | 6.57 |
| | Cross II | 2020 | 8.44 | 7.57 | 9.28 | 8.00 | 0.05 | −0.16 | 1.28 | 15.93 | 9.94 |
| | | 2021 | 7.88 | 7.47 | 9.16 | 7.67 | 0.45 | 0.65 | 1.48 | 19.35 | 16.23 |
| | Cross III | 2020 | 8.75 | 5.38 | 8.79 | 7.06 | 0.16 | 0.38 | 1.73 | 24.48 | 0.51 |
| | | 2021 | 10.26 | 5.65 | 9.16 | 7.96 | 0.66 | 0.07 | 1.20 | 15.08 | −10.75 |
| | Cross IV | 2020 | 5.38 | 8.52 | 8.23 | 6.95 | −0.56 | 0.64 | 1.29 | 18.52 | −3.31 |
| | | 2021 | 5.65 | 7.70 | 8.18 | 6.68 | −0.08 | −0.07 | 1.51 | 22.57 | 6.29 |
| FL | Cross I | 2020 | 17.35 | 15.50 | 14.06 | 16.43 | 0.11 | −0.24 | −2.37 | −14.42 | −18.99 |
| | | 2021 | 18.43 | 16.77 | 16.04 | 17.60 | 0.62 | 1.12 | −1.56 | -8.89 | −12.99 |
| | Cross II | 2020 | 17.35 | 15.97 | 17.32 | 16.66 | 0.26 | 0.18 | 0.66 | 3.95 | −0.19 |
| | | 2021 | 18.43 | 16.05 | 18.19 | 17.24 | 0.24 | −0.22 | 0.95 | 5.49 | −1.34 |
| | Cross III | 2020 | 14.29 | 9.89 | 14.07 | 12.09 | 0.35 | 0.11 | 1.98 | 16.41 | −1.52 |
| | | 2021 | 15.66 | 11.31 | 16.31 | 13.49 | 1.52 | 5.04 | 2.83 | 20.95 | 4.16 |
| | Cross IV | 2020 | 9.89 | 15.50 | 13.62 | 12.69 | 0.09 | 0.64 | 0.93 | 7.31 | −12.13 |
| | | 2021 | 11.31 | 16.77 | 14.90 | 14.04 | −0.15 | 3.62 | 0.85 | 6.07 | −11.20 |
| FPL | Cross I | 2020 | 19.34 | 20.71 | 17.51 | 20.02 | 0.15 | −0.07 | −2.51 | −12.53 | −15.43 |
| | | 2021 | 18.93 | 17.67 | 18.03 | 18.30 | −0.17 | −0.22 | −0.27 | −1.48 | −4.76 |
| | Cross II | 2020 | 19.34 | 14.39 | 18.84 | 16.86 | 0.15 | 0.08 | 1.97 | 11.71 | −2.58 |
| | | 2021 | 18.93 | 14.59 | 16.22 | 16.76 | 0.44 | 0.34 | −0.54 | −3.24 | −14.34 |
| | Cross III | 2020 | 22.69 | 12.04 | 16.21 | 17.36 | 0.66 | 1.34 | −1.15 | −6.62 | −28.54 |
| | | 2021 | 19.08 | 10.14 | 15.86 | 14.61 | 1.20 | 6.01 | 1.25 | 8.58 | −16.87 |
| | Cross IV | 2020 | 12.04 | 20.71 | 15.65 | 16.37 | −0.01 | 1.94 | −0.72 | −4.41 | −24.43 |
| | | 2021 | 10.14 | 17.67 | 14.63 | 13.90 | 0.53 | 0.55 | 0.72 | 5.19 | −17.23 |
| FW | Cross I | 2020 | 0.60 | 0.51 | 0.46 | 0.56 | 0.30 | 0.45 | −0.10 | −17.89 | −24.25 |
| | | 2021 | 0.62 | 0.72 | 0.63 | 0.67 | 0.24 | 0.67 | −0.04 | −6.18 | −12.91 |
| | Cross II | 2020 | 0.60 | 0.51 | 0.83 | 0.56 | 0.06 | −0.47 | 0.28 | 49.38 | 38.14 |
| | | 2021 | 0.62 | 0.67 | 1.00 | 0.64 | 3.94 | 35.21 | 0.36 | 55.12 | 48.78 |
| | Cross III | 2020 | 0.57 | 0.22 | 0.63 | 0.39 | 0.57 | 0.76 | 0.24 | 60.70 | 11.94 |
| | | 2021 | 0.90 | 0.20 | 0.76 | 0.55 | 0.45 | 0.32 | 0.21 | 37.21 | −15.79 |
| | Cross IV | 2020 | 0.22 | 0.51 | 0.51 | 0.37 | 0.19 | 0.10 | 0.14 | 39.08 | 0.00 |
| | | 2021 | 0.20 | 0.72 | 0.58 | 0.46 | 0.28 | 0.38 | 0.12 | 26.03 | −19.08 |
| FI | Cross I | 2020 | 2.08 | 1.81 | 1.83 | 1.94 | 0.33 | −0.09 | −0.12 | −6.02 | −12.14 |
| | | 2021 | 2.38 | 2.18 | 1.94 | 2.28 | 0.45 | 0.15 | −0.33 | −14.66 | −18.22 |
| | Cross II | 2020 | 2.08 | 2.14 | 1.88 | 2.11 | 0.42 | 0.24 | −0.23 | −11.12 | −12.46 |
| | | 2021 | 2.38 | 2.17 | 2.01 | 2.27 | 0.34 | −0.24 | −0.26 | −11.45 | −15.36 |
| | Cross III | 2020 | 1.65 | 1.85 | 1.61 | 1.75 | 0.15 | −0.27 | −0.14 | −8.19 | −13.03 |
| | | 2021 | 1.55 | 2.01 | 1.79 | 1.78 | 0.84 | 1.76 | 0.01 | 0.63 | −10.76 |
| | Cross IV | 2020 | 1.85 | 1.81 | 1.66 | 1.83 | 0.42 | 0.45 | −0.17 | −9.26 | −10.19 |
| | | 2021 | 2.01 | 2.18 | 1.83 | 2.09 | 0.54 | 1.28 | −0.26 | −12.30 | −15.75 |

Note: $P_1$, female; $P_2$, male; MP, the pro-median value; $H_m$, mid-parent heterosis; $RH_m$, ratio of mid-parent heterosis; $RH_b$, ratio of heterobeltiosis.

The ratio of heterobeltiosis ($RH_b$) for FPL and FI in the four crosses were all negative, and the $RH_b$% of FL in CI, CII and CIV groups were less than 0, indicating that these two traits tended to be low-value parental phenotypes in the $F_1$ hybrids. However, in CII, CIII and CIV, the heterobeltiosis is positive for FW, and therefore, this trait tends to be a high-value parental phenotype in hybrids. Therefore, the phenotypic values of five fruit

traits in $F_1$ offspring were exceeded by that of two parents. However, the average value of some offspring traits was between the parents, which indicated that the hybrid offspring generally showed a superparental separation phenomenon.

### 3.3. The Major Gene and Polygene Mixed Inheritance Model for Fruit Traits

The five fruit traits were segregated in four $F_1$ populations (Figure S1). The frequency distribution of the five traits showed good continuity of the multipeak or skewed distribution, with obvious quantitative characteristics, which fits with the main gene plus polygene mixed model.

The Akaike's information criterion (AIC) value and fitness test (Tables S1 and S2) for five fruit traits were calculated and analyzed according to the mixed inheritance model of major gene plus polygene for quantitative traits in a single-generation segregation method. The optimal inheritance model of these five fruit traits was finally determined by the minimum AIC value criterion (Table 5). The results showed that the optimal inheritance models for five fruit traits of CI in 2020 were all A-0, indicating that the inheritance of these five traits was controlled by polygenes, and therefore, greatly affected by the environment. The optimal inheritance models for FD, FL and FI of CI in 2021 were all A-1, indicating that the inheritance of these three traits was controlled by a pair of additive-dominant major genes. In 2020, the optimal inheritance models for FL, FPL and FI of CII were all A-1, and the optimal model for FD, FL and FI in 2021 was also A-1, for which the controlling factor is a pair of additive-dominant major genes. The optimal models for FD, FL and FI of CIII in 2020 were A-0, which is the control model of multi-genes, while the optimal model for these three traits in 2021 was A-1, which belongs to the model for a pair of additive-dominant major gene inheritance. The optimal model for FD of CIV in 2020 and 2021 was B-6, which describes inheritance by two pairs of additive-dominant allele genes. The optimal super parental inheritance model of FI, A-1, was also maintained over the two years, indicative of its inheritance control by a pair of additive-dominant major genes. Therefore, the inheritance of five fruit traits was more unstable in 2020, which may be greatly affected by the environment. In 2021, the inheritance of fruit traits was more stable, indicating their control by one or two pairs of major genes.

**Table 5.** Optimum genetic model for five main fruit traits.

| Traits | Cross I | | Cross II | | Cross III | | Cross IV | |
|---|---|---|---|---|---|---|---|---|
| | **2020** | **2021** | **2020** | **2021** | **2020** | **2021** | **2020** | **2021** |
| FD | A-0 | A-1 | A-0 | A-1 | A-0 | A-1 | B-6 | B-6 |
| FL | A-0 | A-1 | A-1 | A-1 | A-0 | A-1 | A-0 | A-0 |
| FPL | A-0 | B-6 | A-1 | A-1 | A-1 | B-1 | A-0 | A-1 |
| FW | A-0 | A-0 | A-0 | B-1 | A-1 | A-1 | A-0 | A-1 |
| FI | A-0 | A-1 | A-1 | A-1 | A-0 | A-1 | A-1 | A-1 |

Note: A-0, Polygenes; A-1, A pair of additive-dominant major genes; B-1, Two pairs of additive-dominant-epistatic major genes; B-6, Two pairs of equally dominant major genes. Same as Tables 4 and 5 below.

In summary, the optimal inheritance models for FD, FL and FI of CI, CII and CIII in 2021 were A-1, indicating that the inheritance model for these three traits is the same. In addition, the correlation between FD, FL and FI reached a very significant level, so it was inferred that the inheritance model of wolfberry fruit traits may be controlled by a pair of additive-dominant major genes.

### 3.4. The Estimation of Genetic Parameters in the Optimal Inheritance Model for Fruit Traits

The first-order and second-order parameters for fruit traits were estimated from their optimal inheritance models (Table 6). The heredity of FD was controlled by a pair of additive-dominant major genes in CI, CII and CIII, and the additive effects ($d_a$) were 0.56, 0.57, and 0.81, respectively, the dominant effects ($h_a$) were −0.55, −0.57, and −0.80, respectively, and the dominant degree ($h_a/d_a$) was −1.00, indicating that the major genes

were negative partial dominance and overdominance. The heritability of the major genes was 35.03%, 37.10% and 56.94%, respectively. Similarly, the heredity of these three crosses for FL were also controlled by a pair of additive-dominant major genes. The $d_a$ values of the gene pair were 1.25, 2.38 and 1.25, respectively. The $h_a$ values were $-1.25$, $-0.89$ and $-1.25$, respectively, and the $h_a/d_a$ values were $-1.00$, $-0.38$ and $-1.00$, respectively, indicating that the major genes are negative partial dominant and overdominance. The heritability of the major gene pair was 29.66%, 51.29% and 27.97%, respectively. Likewise, the inheritance of FI also is modeled to a pair of additive-dominant major genes, with $d_a$ values of 0.19, 0.23 and 0.18, $h_a$ values of $-0.19$, $-0.22$ and $-0.18$ and $h_a/d_a$ values of $-1.00$, $-0.97$ and $-1.00$, respectively. The results indicate that the major gene pair are negative partial dominant and super dominant, and the heritability of the major gene pair was 42.74%, 47.10% and 50.03%, respectively. In addition, in CIV, the heredity of FD was controlled by two pairs of additive-dominant allelic genes, with a heritability of the major genes of 16.57%, while the inheritance of FL fitted best with the polygenes model. The inheritance of FI in CIV is similar to that observed in CI, CII and CIII, in that it is fitted to a model for a pair of additive-dominant major gene pair with values for $d_a$, $h_a$ and $h_a/d_a$ of 0.13, $-0.13$ and $-1.00$, respectively. The major gene pair is negative hyperdominant and the heritability was 38.48%. In summary, the optimal inheritance model for the five fruit traits in wolfberry is A-1, and its major gene heritability ranges from 25.62% to 56.94%.

**Table 6.** The main gene heritability of five fruit traits in $F_1$ hybrids from four crosses.

| Traits | Combination | Years | Model | First-Order Parameters | | | | | | | | | | Second-Order Parameters | | |
|---|---|---|---|---|---|---|---|---|---|---|---|---|---|---|---|---|
| | | | | $d_a$(d) | $d_b$ | $h_a$(h) | $h_b$ | i | $j_{ab}$ | $j_{ba}$ | l | $h_a/d_a$ | $h_b/d_b$ | $\sigma2_{mg}$ | $\sigma2_p$ | $h2_{mg}$(%) |
| FD | Cross I | 2021 | A-1 | 0.56 | | −0.55 | | | | | | −1.00 | | 0.23 | 0.66 | 35.03 |
| | Cross II | 2021 | A-1 | 0.57 | | −0.57 | | | | | | −1.00 | | 0.24 | 0.65 | 37.10 |
| | Cross III | 2021 | A-1 | 0.81 | | −0.80 | | | | | | −0.99 | | 0.48 | 0.85 | 56.94 |
| | Cross IV | 2020 | B-6 | 0.42 | | | | | | | | | | 0.27 | 0.52 | 52.11 |
| | | 2021 | B-6 | 0.23 | | | | | | | | | | 0.08 | 0.50 | 16.57 |
| FL | Cross I | 2021 | A-1 | 1.25 | | −1.25 | | | | | | −1.00 | | 1.17 | 3.93 | 29.66 |
| | Cross II | 2020 | A-1 | 1.58 | | −1.57 | | | | | | −0.99 | | 1.86 | 6.24 | 29.82 |
| | | 2021 | A-1 | 2.38 | | −0.89 | | | | | | −0.38 | | 3.02 | 5.89 | 51.29 |
| | Cross III | 2021 | A-1 | 1.25 | | −1.25 | | | | | | −1.00 | | 1.18 | 4.22 | 27.97 |
| FPL | Cross I | 2021 | B-6 | 1.31 | | | | | | | | | | 2.56 | 6.34 | 40.43 |
| | Cross II | 2020 | A-1 | 2.16 | | −2.09 | | | | | | −0.97 | | 3.43 | 6.24 | 54.92 |
| | | 2021 | A-1 | 1.76 | | −1.75 | | | | | | −1.00 | | 2.31 | 5.50 | 41.98 |
| | Cross III | 2020 | A-1 | 1.57 | | −1.57 | | | | | | −1.00 | | 1.84 | 5.29 | 34.75 |
| | | 2021 | B-1 | 1.52 | 1.50 | −2.25 | −1.48 | 1.49 | −1.48 | −0.73 | 2.24 | −1.50 | −0.99 | 2.50 | 4.41 | 56.65 |
| | Cross IV | 2021 | A-1 | 1.37 | | −1.37 | | | | | | −1.00 | | 1.41 | 3.53 | 39.84 |
| FW | Cross II | 2021 | B-1 | 0.61 | 0.61 | −0.91 | −0.61 | 0.61 | −0.61 | −0.30 | 0.91 | −1.50 | −1.00 | 0.41 | 0.08 | 494.77 |
| | Cross III | 2020 | A-1 | 0.09 | | −0.09 | | | | | | −1.00 | | 0.01 | 0.02 | 32.56 |
| | | 2021 | A-1 | 0.11 | | −0.11 | | | | | | −0.99 | | 0.01 | 0.02 | 40.14 |
| | Cross IV | 2021 | A-1 | 0.07 | | −0.07 | | | | | | −1.00 | | 0.00 | 0.01 | 25.62 |
| FI | Cross I | 2021 | A-1 | 0.19 | | −0.19 | | | | | | −1.00 | | 0.03 | 0.06 | 42.74 |
| | Cross II | 2020 | A-1 | 0.19 | | −0.19 | | | | | | −0.99 | | 0.03 | 0.07 | 40.58 |
| | | 2021 | A-1 | 0.23 | | −0.22 | | | | | | −0.97 | | 0.04 | 0.08 | 47.10 |
| | Cross III | 2021 | A-1 | 0.18 | | −0.18 | | | | | | −1.00 | | 0.02 | 0.05 | 50.03 |
| | Cross IV | 2020 | A-1 | 0.10 | | −0.10 | | | | | | −1.00 | | 0.01 | 0.02 | 34.59 |
| | | 2021 | A-1 | 0.13 | | −0.13 | | | | | | −1.00 | | 0.01 | 0.03 | 38.48 |

Note: $d_a$, the first major gene additive effect; $d_b$, the second major gene additive effect; $h_a$, the first major gene dominant effect; $h_b$, the second major gene dominant effect; $\sigma^2_p$, phenotypic variance; $\sigma^2_{mg}$, major gene variance; $h^2_{mg}$, major gene heritability.

## 4. Discussion

Controlling plant phenotypic characteristics is usually one of the most important goals in plant genetic breeding [24–27]. The fruit traits are important economic traits and targets for genetic improvement [28]. Achieving the characteristics preferred by consumers in wolfberry varieties will accelerate the development of the wolfberry industry. Current research in the analysis of wolfberry genetics includes SNP-based high-density genetic mapping of leaf and fruit-related quantitative trait loci in Lycium Linn [29], hybrid identification and genetic analysis in a wolfberry $F_1$ population using SSR markers [21], and an SSR marker-based study of the genetic diversity in 24 germplasm resources of *Lycium bararum L.* [30]. However, wolfberry genetic studies have been based on phenotypic characteristics [31–33], and the research on the hereditary law of fruit traits lags behind, with very few existing reports. In this study, it was found that the FI trait in wolfberry had a highly significant positive correlation with FL, and a very significant negative correlation with FD. Moreover, FI displayed the largest correlation coefficient with FL, compared with the other three traits (Table S3). The wolfberry fruit type largely depends on the change of FL, which is consistent with the conclusion of Zhang et al. [28]. Heterosis is an important breeding strategy in many plants [34]. Wolfberry is a cross-pollinating plant and its genetic material is highly heterozygous [21], so that in general, observed heterosis can be attributed mainly to the heterogeneity between the parents [11]. In this study, the analysis of five wolfberry fruit traits indicated that FD, FL and FW in CII, CIII and CIV showed positive dominance, with $RH_m$% values ranging from 3.95% to 60.70% (Table 4), indicating that these three traits have significant heterosis, which is in agreement to other reports [8,11,14]. The existence of the heterosis may be due to the high heterozygosity in wolfberry. The phenotypic values of five fruit traits exceeded their parents in four crosses, which further indicates that the hybrid offspring in wolfberry generally has the superparental separation phenomenon, showing rich genetic diversities in these traits. Therefore, it is possible to select individual plants displaying excellent fruit shape and other characteristics from the $F_1$ population and then maintain these characteristics through vegetative propagation (cloning), which would significantly accelerate the exploitation of the desired wolfberry characteristics in agriculture.

In this paper, the mixed major gene plus polygene inheritance model for five fruit traits in wolfberry was used to investigate a single segregation generation. The results of the A-1, A-0, B-6 and B-1 models indicated that the data were best fitted to the A-1 inheritance model (Table 5). The heredity of FD, FL and FI was controlled by a pair of additive-dominant major genes in CI, CII and CIII, and the major gene heritability ($h^2_{mg}$) is 25.62–56.94%, which is a medium heritability (Table 6). In addition, the heredity of FPL in CII and FPL and FW in CIV are also controlled by a pair of additive-dominant major genes. Previous studies have shown that plant phenotypic traits are the result of the close interaction between genes and environment [35,36]. Thus, we can infer that the heredity of the five fruit traits in wolfberry is mainly controlled by major genes and environmental conditions. Moreover, the additive effect of the major genes of FL in CII is high ($d_a = 2.38$), reflecting that this trait had a good breed value in wolfberry [11]. In addition, the heredity of FW in CII and FPL in CIII was controlled by two pairs of additive-dominant-epistatic major genes. Based on the B-1 model, the additive plus dominant effects (j) of the two genes for FW and FPL is calculated as negative, suggesting that there existed barriers to select such traits in early generations. However, the additive x additive effects (i) of the two genes were positive, indicating that these traits could be possibly selected in late generations [37]. Simultaneously, the heritability of the FW in CII is >1; thus, we can safely presume that the interaction effects of the two major genes controlling this trait are strong. During wolfberry breeding programs, full advantage should be taken of the two main gene combinations and interaction effects to improve the selection of fruit traits.

In the present study, the number of genes controlling the five fruit traits in wolfberry was also determined through the separation analysis method of the $F_1$ population (a pseudo-$F_2$ population). These traits were both controlled by polygenes (A-0), a pair of

additive-dominant major genes (A-1) and controlled by two pairs of additive-dominant-epistatic major genes (B-1) and two pairs of additive-dominant alleles (B-6) (Table 4). These results can provide some reference for molecular marker-assisted breeding of the main fruit characteristics in wolfberry, which is highly consistent with the stable fruit QTL loci obtained by Zhao et al. [29] using a high-density genetic map constructed from a $F_1$ population. However, based on phenotypic data, single-generation segregation analysis methods can only detect and estimate the genetic parameters of major genes, and cannot distinguish the relationship between polygenic and environmental variations. Therefore, in order to better explore the spatiotemporal expression of quantitative genes, subsequent studies should use joint segregation analysis methods to accurately measure traits over several years, different locations and multiple generations.

## 5. Conclusions

To reveal the genetic mechanism of the wolfberry fruit, this study used a multivariate statistical analysis and hybrid genetic analysis to analyze five fruit-related traits in four $F_1$ populations. The obvious phenotypic variations and extremely significant differences among the four different hybrid combinations revealed that these traits were largely influenced by environment and quite separated in hybrid offspring. Mixed genetic analysis demonstrated that the five traits in different environment showed diverse genetic control mechanisms, and the inheritance of these traits was controlled by both environment and genes. The results of this study laid a foundation for elucidating the genetic mechanism of wolfberry and further carrying out the genetic breeding of wolfberry.

**Supplementary Materials:** The following supporting information can be downloaded at: https://www.mdpi.com/article/10.3390/agronomy12061403/s1, Table S1: AIC values of the $F_1$ population from four crosses under different genetic models; Table S2: Suitability test of the selected model from four crosses; Table S3: The correlation of the main fruit traits in the $F_1$ population from four crosses. Figure S1: The main gene heritability of five main fruit traits in $F_1$ hybrids from four crosses. (a) The frequency distribution for fruit traits in $F_1$ population derived from the cross I; (b) The frequency distribution for fruit traits in $F_1$ population derived from the cross II; (c) The frequency distribution for fruit traits in $F_1$ population derived from the cross III; (d) The frequency distribution for fruit traits in $F_1$ population derived from the cross IV.

**Author Contributions:** Conceptualization, X.R. and H.L.; methodology, T.H. and B.Z.; software, X.R.; validation, J.Z. and W.A.; formal analysis, J.Z. and W.A.; investigation, Y.W., H.L., X.L. R.W., Y.Y. and L.D.; data curation, X.R. and H.L.; writing—original draft preparation, X.R., J.Z. and H.L.; writing—review and editing, J.Z. and W.X.; visualization, J.Z. All authors have read and agreed to the published version of the manuscript.

**Funding:** This research was sponsored by the Ningxia Hui Autonomous Region Science and Technology Innovation Leading Talents Project (KJT2017004), the Key Research & Development Program of Ningxia Hui Autonomous Region (2021BEF02002), the Innovative Research Group Project of Ningxia Hui Autonomous Region (2021AAC01001) and the National Natural Science Foundation of China (32060359).

**Institutional Review Board Statement:** Not applicable.

**Informed Consent Statement:** Not applicable.

**Data Availability Statement:** The data obtained in preparing this manuscript are available from the authors upon request.

**Acknowledgments:** The authors would like to thank Yuhui Xu for his technical assistance for editing this manuscript.

**Conflicts of Interest:** The authors declare no conflict of interest.

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
