# Peer review of "Genetic Analysis of Fruit Traits in Wolfberry (Lycium L.) by the Major Gene Plus Polygene Model"

_agronomy, doi:10.3390/agronomy12061403_

Round 1
Reviewer 1 Report
The present research is interesting. The topic of the paper is relevant, timely, and of interest to the audience of the journal. A lot of work has been done by the authors and generally the manuscript is well written.
However, I have to note the following:
1. The authors have to give a brief description of the models in a table. It will be easier for the reader to follow the text
2. The authors have to rewrite the conclusions section or to delete it. In its present form does not provide anything extra to the manuscript. Some results are just repeated and in many places it is the same as the abstract
3. My suggestions are indicated in the accompanying document

Author Response
Dear Editor and Reviewers,
Many thanks for handling with our manuscript entitled “Genetic Analysis of Fruit Traits in Wolfberry (Lycium L.) by the Major Gene Plus Polygene Model”. The comments were all valuable and very helpful for revising and improving our paper. According to editor and the reviewers’ comments and suggestions, we have modified the text in this revised version. Most of the suggested edits have been applied, and we revised some sentences/content to ensure a more precise presentation of our work (red font). The detailed responses to the reviewers’ comments are attached. We hope the current manuscript is now suitable for publication.
This revised manuscript has been seen and approved by all listed authors. The authors have declared that no competing interests exist. If you have any questions about this paper, please don’t hesitate to let me know.
Looking forward to hearing from you.
Best regards,
Jianhua Zhao
National Wolfberry Engineering Research Center/Wolfberry Science Research Institute, Ningxia Academy of Agriculture and Forestry Sciences, Yinchuan, 750002, China
Reviewer 1
The present research is interesting. The topic of the paper is relevant, timely, and of interest to the audience of the journal. A lot of work has been done by the authors and generally the manuscript is well written.
However, I have to note the following:
- The authors have to give a brief description of the models in a table. It will be easier for the reader to follow the text
Response:Thanks. The following description table (Table 1 in the MS) has been made in line 170 to 173.
|
Table 1. Descriptions of the genetic models |
|||
|
Model implication |
Model |
Model description |
|
|
|
|||
|
0MG |
A-0 |
Polygenes |
|
|
1MG-AD |
A-1 |
A pair of additive-dominant major genes |
|
|
1MG-A |
A-2 |
A pair of additive major genes |
|
|
1MG-EAD |
A-3 |
A pair of fully dominant major genes |
|
|
1MG-AEND |
A-4 |
A pair of negative fully dominant major genes |
|
|
2MG-ADI |
B-1 |
Two pairs of additive-dominant-epistatic major genes |
|
|
2MG-AD |
B-2 |
Two pairs of additive-dominant major genes |
|
|
2MG-A |
B-3 |
Two pairs of additive major genes |
|
|
2MG-EA |
B-4 |
Two pairs of equally additive major genes |
|
|
2MG-AED |
B-5 |
Two pairs of fully dominant major genes |
|
|
2MG-EEAD |
B-6 |
Two pairs of equally dominant major genes. |
|
|
Note: MG represents the major inheritance model; A represents the additive effect; D represents the dominance effect; E represents equal; N represents negative and I represents epistatic interaction. |
|
||
- The authors have to rewrite the conclusions section or to delete it. In its present form does not provide anything extra to the manuscript. Some results are just repeated and in many places it is the same as the abstract.
Response: Thanks for this comment. The conclusion has been rewritten in line 367 to 376 as the following: To reveal the genetic mechanism of wolfberry fruit, this study used a multivariate statistical analysis and hybrid genetic analysis to analyze five fruit-related traits in four F1 populations. The obvious phenotypic variations and extremely significant differences among the four different hybrid combinations revealed that these traits were largely influenced by environment and quite separated in hybrid offsprings. Mixed genetic analysis demonstrated that the five traits in different environment showed a diverse genetic control mechanisms, and the inheritance of these traits was controlled by both environment and genes. The results of this study laid a foundation for elucidating the genetic mechanism of wolfberry and further carrying out the genetic breeding of wolfberry.
- My suggestions are indicated in the accompanying document.
Response: Thanks. We have modified the MS according to your suggestions.

Reviewer 2 Report
In the study, the researchers investigated the genetic inheritance of 5 fruit traits in wolfberry using four F1 populations. The result showed that FD, FPL, FW and FI were affected by both genes and the environment. This is very important for future breeding. I have the following question:
(1) In the Introduction section, it should be explained in detail that how many models were existed in the mixed inheritance analysis, such as D-2, E-1, E-0, D-0, and B-2.
(2) The fonts in Figure 1 are too few and should be no less than 6 pt.
(3) Please added some method of phenotypic evaluation although there are references 22.
(4) The “years” in Table 1 can be modified to “Populations”.
(5) Figure 2 should be replaced as an attachment?
Author Response
Dear Editor and Reviewers,
Many thanks for handling with our manuscript entitled “Genetic Analysis of Fruit Traits in Wolfberry (Lycium L.) by the Major Gene Plus Polygene Model”. The comments were all valuable and very helpful for revising and improving our paper. According to editor and the reviewers’ comments and suggestions, we have modified the text in this revised version. Most of the suggested edits have been applied, and we revised some sentences/content to ensure a more precise presentation of our work (red font). The detailed responses to the reviewers’ comments are attached. We hope the current manuscript is now suitable for publication.
This revised manuscript has been seen and approved by all listed authors. The authors have declared that no competing interests exist. If you have any questions about this paper, please don’t hesitate to let me know.
Looking forward to hearing from you.
Best regards,
Jianhua Zhao
National Wolfberry Engineering Research Center/Wolfberry Science Research Institute, Ningxia Academy of Agriculture and Forestry Sciences, Yinchuan, 750002, China
Reviewer 2
In the study, the researchers investigated the genetic inheritance of 5 fruit traits in wolfberry using four F1 populations. The result showed that FD, FPL, FW and FI were affected by both genes and the environment. This is very important for future breeding. I have the following question:
- In the Introduction section, it should be explained in detail that how many models were existed in the mixed inheritance analysis, such as D-2, E-1, E-0, D-0, and B-2.
Response: Genetic models have been added in line 43 to line 50.
- The fonts in Figure 1 are too few and should be no less than 6 pt.
Response: Thanks for this suggestion. We have modified Figure 1 as your recommendation.
- Please added some method of phenotypic evaluation although there are references 22.
Response: We have added some details of phenotyping in line 135 to 138.
(4) The “years” in Table 1 can be modified to “Populations”.
Response: Fixed.
(5) Figure 2 should be replaced as an attachment?
Response: Figure 2 has been replaced with an attachment (Figure S1).
